# Estimating the time-varying reproduction number of COVID-19 with a state-space method

**Shinsuke Koyama[1], Taiki Horie[2], Shigeru Shinomoto[2,3]***

**1** The Institute of Statistical Mathematics, Tokyo, Japan, **2** Department of Physics, Kyoto University, Kyoto, Japan, **3** Brain Information Communication Research Laboratory Group, ATR Institute International, Kyoto, Japan

* shinomoto.shigeru.6e@kyoto-u.ac.jp

**Data Availability Statement:** The application program and example datasets are available at our website https://s-shinomoto.com/COVID/ and the site hosted publicly on GitHub, accessible via https://github.com/shigerushinomoto.

## Abstract

After slowing down the spread of the novel coronavirus COVID-19, many countries have started to relax their confinement measures in the face of critical damage to socioeconomic structures. At this stage, it is desirable to monitor the degree to which political measures or social affairs have exerted influence on the spread of disease. Though it is difficult to trace back individual transmission of infections whose incubation periods are long and highly variable, estimating the average spreading rate is possible if a proper mathematical model can be devised to analyze daily event-occurrences. To render an accurate assessment, we have devised a state-space method for fitting a discrete-time variant of the Hawkes process to a given dataset of daily confirmed cases. The proposed method detects changes occurring in each country and assesses the impact of social events in terms of the temporally varying reproduction number, which corresponds to the average number of cases directly caused by a single infected case. Moreover, the proposed method can be used to predict the possible consequences of alternative political measures. This information can serve as a reference for behavioral guidelines that should be adopted according to the varying risk of infection.

## Author summary

Society and the media alternate between hope and despair in response to the temporary decrease or increase of daily new COVID-19 infections. The number of cases has been dependent on the political measures that were adopted in each country. Accordingly, there is a strong demand for quantifying the effects of individual measures. The reproduction number, defined as the average number of cases directly caused by a single infected case, is one of the indices of the current infectivity status. To capture the time-varying reproduction number correctly, it is necessary to incorporate the distribution of delays, which are widely dispersed from 2 to 14 days for the case of COVID-19. We have developed a state-space method for estimating the reproduction number solely from an available dataset of the number of daily cases. Our method automatically detects the change-

**Funding:** This work has been funded by New Energy and Industrial Technology Development Organization (NEDO) (to SS). The funders had no role in study design, data collection and analysis, decision to publish, or preparation of the manuscript.

**Competing interests:** The authors have declared that no competing interests exist.

points in the reproduction number. We apply our method to the real data and examine if the detected changes are consistent with the times at which political measures had been taken in each country. Furthermore, our method can be used to predict the number of new cases in the future to examine the possible consequences of alternative political measures.

## Introduction

While the novel coronavirus COVID-19 has spread worldwide, different countries have employed various intervention strategies, many of which were later followed by liberalized approaches and relaxed behaviour from individuals. In this situation, it is desirable to monitor the extent to which individual political measures have influenced the spread of the disease in each country and predict the possible consequences of alternative measures.

A fundamental metric representing the degree of the spread of a disease is the reproduction number, $R$, which is defined as the average number of cases directly caused by a single case [1, 2]. However, it is difficult to trace the concrete processes by which infections have been transmitted among individual people, particularly considering the protection of private information. Thus, a statistical analytical method is needed to infer the underlying process from the available data consisting of the number of daily infected cases, which were obtained by imperfect observation and accompanied by errors.

Mathematical epidemiological studies using the ordinary differential equation (ODE) models, such as the susceptible–infectious–recovered (SIR) model, have contributed to our understanding of causal factor dynamics, the results of which can be used to suggest control measures needed in given situations [3–5]. While the original study of Kermack and McKendrick in 1927 [6] considered the distribution of delays in the transmission of a disease, the majority of later studies used ODEs in favor of an analytical treatment [7]. Though ODE models also assume the transmission delay, such that the SIR model represents the situation in which delays are distributed exponentially [8], they cannot adopt the specific distribution of delays for each disease. In the case of COVID-19, transmission delays are known to be widely dispersed from 2 to 14 days [9–12]. To capture the time-varying reproduction number under fluctuating circumstances, it is necessary to incorporate the delay distribution explicitly in the analysis, as previously performed in an analysis using the semi-mechanistic Bayesian hierarchical model [13].

Recently, Chiang, Liu, and Mohler [14] modeled COVID-19 transmission using the Hawkes process [15], in which the delay distribution can be explicitly adopted as a self-exciting kernel. They combined the Hawkes process with spatial and temporal covariates, such as demographic features and Google mobility indices, to explain the variability of the reproduction number, and to forecast future cases and deaths in the USA.

Herein, we establish a state-space method for estimating the time-varying reproduction number by fitting a discrete-time variant of the Hawkes process. While the semi-mechanistic Bayesian hierarchical model [13] requires manual assignment of the change-points, our method automatically detects the change-points solely from a given series of the number of daily cases. We first apply the method to synthetic data to confirm that the method properly detects the change-points embedded in the simulation. Here, the proposed method is compared with a conventional method in terms of performance estimation of the time-varying reproduction ratio. Then, we apply the proposed method to real data and examine whether the detected changes are consistent with the times at which political measures had been

implemented in each country. The proposed method can also predict the number of new cases in the future to examine any possible consequences of alternative political measures.

## Methods

We have developed a state-space method of estimating the temporally changing reproduction number from a given series of the number of daily new infections, by introducing a discrete-time variant of the Hawkes process as a basic model describing the transmission of disease. The state-space method describes the evolution of a system by a set of first-order difference equations of state variables. The state variables can be inferred from measured data using a recursive Bayesian estimation [16].

### The rate process on a daily basis

The (original, continuous-time) Hawkes process describes a self-excitation process in terms of the instantaneous occurrence rate $\lambda(t)$ as

$$\lambda(t) = \mu + R\sum_{t_k < t}\phi(t - t_k),\tag{1}$$

where $\mu$ is the spontaneous occurrence rate, and the second term represents a self-excitation effect such that the occurrence of an event adds the probability of future events (Fig 1). $R$ is the reproduction number representing the average number of events induced by a single event, $t_k$ is the occurrence time of a past ($k$th) event, and $\phi(t)$ is a kernel representing the distribution of the transmission delays, satisfying the normalization $\int_0^\infty \phi(t)dt = 1$. Events $\{t_1, t_2, \ldots\}$ are derived randomly in time from the rate $\lambda(t)$.

For the case of COVID-19, however, exact timing of infection event is not available. To deal with the numbers of daily new cases that are practically available, we convert the original Hawkes process Eq (1) into a discrete-time variant representing the expected number of events on a daily basis:

$$\lambda_j = \mu' + \sum_{i=1}^{j-1}v_iR_i\phi_{j-i},\tag{2}$$

where $\lambda_j$ is the expected number of events on $j$th day. The first term $\mu'$ on the right-hand-side refers to the expected number of spontaneous occurrences on a daily basis. The second term represents the self-excitation process, the manner in which $v_i$ events that have occurred on a

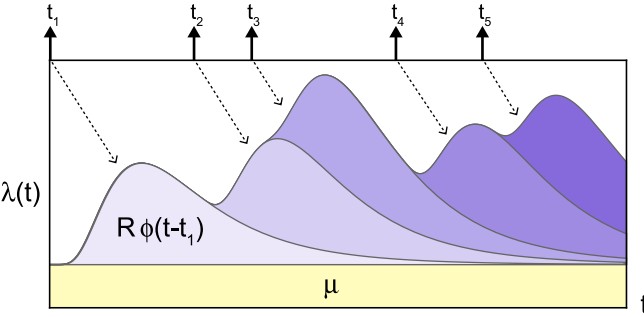

**Fig 1. Schematic description of the Hawkes process Eq (1).** The occurrence rate $\lambda(t)$ is increased according to past events occurred at times $t = t_k$ ($k = 1, 2, \ldots$) with the transmission delays $t - t_k$ distributed with $\phi(t - t_k)$. $R$ is the reproduction number that represents the average number of events induced by a single event.

day $i$ exerted influence with the delay of $j - i$ days. Here, we assume that the reproduction number may change and represent the daily dependence as $\{R_i\}_i$. $\phi_{j-i}$ represents a distribution of the transmission delays $d = j - i$, satisfying the normalization $\sum_{d=1}^{\infty} \phi_d = 1$.

The number of events $v_i$ or $v$ is derived from a distribution specified with the mean rate $\lambda_i$ or $\lambda$. It would be natural to assume the Poisson distribution $p(v|\lambda) = \frac{\lambda^v}{v!} e^{-\lambda}$. However, real data are subject to erroneous observation and accordingly they tend to be over-dispersed, or the sample variance exceeds the sample mean. Here, we incorporate over-dispersed data using the negative binomial distribution in the following form [17]:

$$p(v|\lambda, \rho) = \frac{\Gamma\left(v + \frac{\lambda}{\rho}\right)}{\Gamma(v + 1)\Gamma\left(\frac{\lambda}{\rho}\right)} \left(\frac{\rho}{1 + \rho}\right)^v \left(\frac{1}{1 + \rho}\right)^{\frac{\lambda}{\rho}},$$ 

(3)

where $\rho(> 0)$ represents the degree of over-dispersion, or the variance is $\mathrm{Var}(v) = (1 + \rho)\lambda$. The Poisson distribution is in the limit of $\rho \to 0$.

The COVID-19 model parameters were chosen as follows: the spontaneous occurrence of infection is absent, $\mu' = 0$ because there is no spontaneous occurrence for COVID-19 except at the initial occurrence in China. The virus is transmitted between individuals during close contact, and each individual is determined to have an episode of infection. The duration between symptom onsets of successive cases is referred to as the serial interval [18], which is slightly different from the incubation period [19, 20]. It is reported that the distribution of the serial intervals is suitably approximated with the log-normal distribution function of the mean 4.7 days and SD 2.9 days for COVID-19 [12]. We have adopted this distribution as the transmission delay kernel $\phi_d$. The distribution of transmission delays on a daily basis is given as the difference of a cumulative distribution function of the log-normal distribution, $\phi_d = \Phi_d - \Phi_{d-1}$, where

$$\Phi_d = \frac{1}{2} \mathrm{erfc}\left(-\frac{\log d - \mu}{\sqrt{2\sigma^2}}\right),$$

(4)

where the parameters $\mu$ and $\sigma$ are given in terms of the mean $m = 4.7$ and the SD $s = 2.9$ as $\mu = \log\left(m^2/\sqrt{s^2 + m^2}\right)$ and $\sigma = \sqrt{\log\left(1 + s^2/m^2\right)}$ (Fig 2).

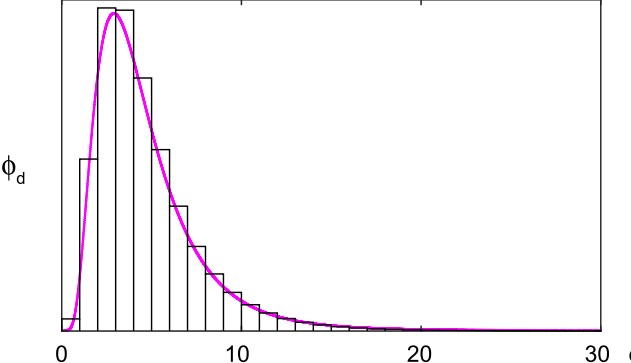

**Fig 2. The distribution of transmission delays.** A bar histogram represents the distribution of transmission delays on a daily bases $\phi_d$, which was converted from the log-normal distribution with the mean 4.7 days and SD 2.9 days (a magenta line).

## The system equation

To detect change-points in the reproduction number $\{R_i\}_i$ in Eq (2), we introduce a method of estimating stepwise dynamics [21]. We assume that system's state $x_i$ obeys the evolution

$$x_i = x_{i-1} + \xi_i, \tag{5}$$

with the Cauchy random number $\xi$:

$$p(\xi|\gamma) = \frac{\gamma}{\pi(\xi^2 + \gamma^2)}. \tag{6}$$

We assume that the reproduction number is given as $R_i = f(x_i)$ with the non-negative function. Here we adopted a ramp function $f(x) = \max(0, x)$.

## State inference

We constructed a state-space model for estimating the temporally changing reproduction number $R_i$ from a given dataset of daily confirmed cases $\{v_1, \ldots, v_T\}$. The basic procedure of constructing the state-space method is similar to the one we developed for estimating exogenous and endogenous factors in a chain of point events [22].

To put the model in the state-space form, we take the summation in Eq (2) over the last $L$ days,

$$\lambda_j = \mu' + \sum_{i=j-L}^{j-1} v_i f(x_i) \phi_{j-i}, \tag{7}$$

and introduce a concatenated state vector,

$$X_i := (x_{i-1}, \ldots, x_{i-L})^{\mathrm{T}}, \tag{8}$$

so that the rate process (7) depends only on the current state $X_i$. Accordingly, the state $X_i$ obeys the evolution

$$X_i = FX_{i-1} + G\xi_{i-1}, \tag{9}$$

where

$$F = \begin{pmatrix} 1 & 0 & \ldots & 0 \\ 1 & 0 & \ddots & \vdots \\ \vdots & \ddots & \ddots & 0 \\ 0 & \ldots & 1 & 0 \end{pmatrix}, \quad G = \begin{pmatrix} 1 \\ 0 \\ \vdots \\ 0 \end{pmatrix}. \tag{10}$$

We have chosen $L = 30$ in the following analysis because the transmission delay kernel $\phi_d$ is negligible at $d = 30$ (Fig 2).

The posterior distribution of system's state $X_i$, given a set of daily new cases until $i$th day $Y_i := \{v_1, \ldots, v_i\}$ is obtained using Bayes' theorem as

$$p(X_i|Y_i) = \frac{p(v_i|X_i)p(X_i|Y_{i-1})}{p(v_i|Y_{i-1})}. \tag{11}$$

Here, $p(X_i|Y_{i-1})$ may be obtained using a system model $p(X_i|X_{i-1})$ and the posterior

distribution on day $i - 1$, $p(X_{i-1}|Y_{i-1})$, as

$$p(X_i|Y_{i-1}) = \int p(X_i|X_{i-1})p(X_{i-1}|Y_{i-1})dX_{i-1}. \tag{12}$$

Starting from the initial distribution $p(X_1|Y_0)$, we iterate Eqs (11) and (12) to compute $p(X_i|Y_{i-1})$ and $p(X_i|Y_i)$ for $i = 1, 2, \ldots, T$.

Then, we compute the distribution of system's states $\{X_1, \ldots, X_T\}$, given an entire set of occurrences $Y_T := \{v_1, \ldots, v_T\}$ with

$$p(X_i|Y_T) = p(X_i|Y_i) \int \frac{p(X_{i+1}|Y_T)p(X_{i+1}|X_i)}{p(X_{i+1}|Y_i)} dX_{i+1} \tag{13}$$

in reverse order as $i = T - 1, T - 2, \ldots, 1$, using the distribution functions $p(X_i|Y_i)$ and $p(X_i|Y_{i-1})$, which were obtained with Eqs (11) and (12).

We then take the median of the posterior distribution $p(X_{i+1}|Y_T)$ for the estimate of the state $\hat{X}_{i+1}$. The estimate of the reproduction number, $\hat{R}_i$, is then given by the first element of $f(\hat{X}_{i+1})$. With the estimated reproduction number, we obtain the estimated total rate as

$$\hat{\lambda}_j = \mu' + \sum_{i=1}^{j-1} v_i \hat{R}_i \phi_{j-i}. \tag{14}$$

We devised an algorithm that performs the integrations in Eqs (11), (12) and (13) numerically using a sequential Monte Carlo method [23, 24].

To avoid bias in estimating the state, which is caused by outliers in the data, we may discard the preassigned outliers and treat them as "missing observations" [24], for which the posterior distribution of $X_i$, conditional on $Y_i$, is set to $p(X_i|Y_i) = p(X_i|Y_{i-1})$ without applying the Bayesian update (11).

## Results

### Analysis of synthetic data

Firstly, we evaluated the functionality of the state-space method by applying it to synthetic data. For this purpose, we constructed simulations of the Hawkes process mimicking prototypical evolutions in several countries. In the simulations, we took $\mu' = 0$ and began with a few infections as initial seeds, mimicking those who introduced the disease into each country. With an initial reproduction number $R > 1$, the daily cases initially grew exponentially. To reproduce a variety of evolutions in different countries, we have evaluated several schedules of the reproduction number $\{R_i\}_i$.

Fig 3 depicts three prototypical cases: (A) the rapid increase is followed by a slow decrease; (B) the rapid increase is followed by a rapid decrease, and then it started to increase again; (C) the increase is followed by a decrease, and then another large increase. For each type of time-varying reproduction number $\{R_i\}_i$, the Hawkes process was simulated over an interval of length $T = 120$ days to generate daily cases $\{v_1, \ldots, v_T\}$. In the simulation, the parameter of the negative binomial distribution Eq (3) was set to $\rho = 50$.

**Parameters of the state-space method.** For each series of simulated data, we have applied the state-space method, or performed the sequential Monte Carlo algorithm to compute the posterior distributions of the reproduction number for each day, $\hat{R}_i$. Here, the over-dispersion

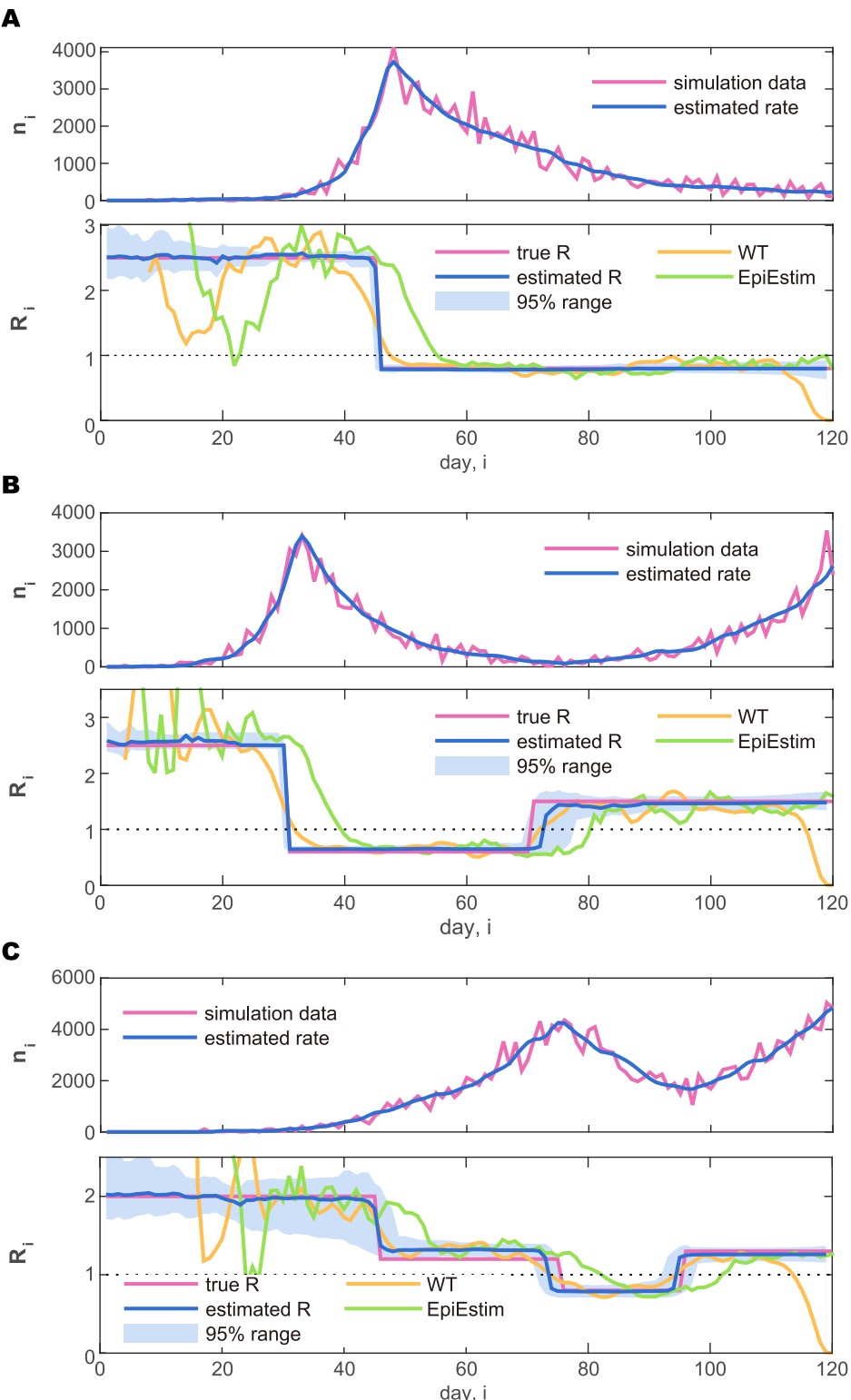

**Fig 3. Synthetic daily cases generated by simulating the Hawkes-type count process and the estimated reproduction number.** (A) Rapid increase followed by a slow decrease. (B) Increase followed by a rapid decrease, and then an increase. (C) Slow increase followed by a decrease, and then another large increase. In the upper panel plotting the number of daily cases (purple line), the rate estimated by the state-space method $\hat{\lambda}_i$ is also plotted (blue line). In the lower panel, the reproduction number $\hat{R}_i$ estimated with the state-space method is plotted in reference to the true

reproduction number $R_i$ (purple line). The blue solid line and the shaded area represent the median and 95% range of the posterior distribution, respectively. The reproduction numbers estimated by Wallinga and Teunis (WT: orange line) and by Cori et al. (EpiEstim: green line) are also plotted for reference.

parameter $\rho$ of the negative binomial distribution Eq (3) is determined from a given dataset as

$$\hat{\rho} = \frac{1}{T} \sum_{i=1}^{T} \frac{(v_i - \bar{\lambda}_i)^2}{\bar{\lambda}_i} - 1, \tag{15}$$

where $\bar{\lambda}_i = \sum_{j=-3}^{3} v_{i+j}/7$ represents the mean daily cases averaged over a week.

To verify the convergence of the posterior estimate of $\hat{R}_i$ concerning the number of particles, we computed the standard error of $\hat{R}_i$ with 100 cases of the Monte Carlo estimation (Table 1). We observed that $10^6$ particles may provide a reasonably accurate estimate of the reproduction number.

The state-space method possesses a hyperparameter $\gamma$ that characterizes the Cauchy distribution of the system equation Eq (6). We tested different values for the hyperparameter $\gamma$ as $10^{-2}$, $10^{-3}$, and $10^{-4}$, and observed that the estimated reproduction number $\hat{R}_i$ is sensitive to the value of $\gamma$ while the estimated total rate $\hat{\lambda}_i$ is robust against $\gamma$ (results not shown). We confirmed that the likelihood was highest for the case of $\gamma = 10^{-3}$. Accordingly, we fixed the hyperparameter at $\gamma = 10^{-3}$ throughout the following analysis.

With the hyperparameter $\gamma = 10^{-3}$ and the parameter $\rho = \hat{\rho}$ determined for each dataset with Eq (15), we performed the sequential Monte Carlo algorithm with $10^6$ particles to compute the posterior distributions of the reproduction number for each day, $\hat{R}_i$. Fig 3 depicts the median (solid line) and 95% range (shaded areas) of the posterior distributions. We see that the amplitude of the reproduction number is estimated properly. In particular, the method has successfully detected change-points in $\{R_i\}_i$ that were embedded in the simulation.

**Comparison with conventional estimation methods.** We compared our method with the following two conventional estimation methods in their ability at estimating the time-varying reproduction number of the synthetic data. A method suggested by Wallinga and Teunis (WT method) estimates the "case reproduction number" [25, 26],

$$R_i = \sum_{j=i+1}^{T} \frac{v_j \phi_{j-i}}{\sum_{k=1}^{j-1} v_k \phi_{j-k}}. \tag{16}$$

Another method suggested by Cori et al. (EpiEstim) estimates the "instantaneous reproduction number" [27, 28],

$$R_i = \frac{v_i}{\sum_{j=1}^{i-1} v_j \phi_{i-j}}, \tag{17}$$

in a Bayesian framework with a Poisson likelihood and a gamma-distributed prior for $R_i$. We

**Table 1. Convergence of the posterior estimate of $\hat{R}_i$.**

| particles | $10^3$ | $10^4$ | $10^5$ | $10^6$ |
|---|---|---|---|---|
| standard error | 0.165 | 0.057 | 0.028 | 0.010 |

The standard error of $\hat{R}_i$ computed with 100 cases of the sequential Monte Carlo algorithm applied to the synthetic data in Fig 3A.

plotted the estimation results obtained by the WT method and EpiEstim with a sliding window of 7 days in Fig 3. It is observed that both methods are easily influenced by the fluctuation of data and accordingly it is difficult to discern the change points in the original process. Furthermore, the WT method tends to underestimate the current reproduction number at the end of the recorded interval, because it requires data that could be obtained in the future; the reproduction number estimated by EpiEstim is shifted backward in time relative to the WT and our methods because it uses only data from time points before $i$.

## Analysis of real data

Next, we applied the state-space method to real data of daily confirmed cases in several countries. The number of daily new cases in various countries is made available on websites hosted by public research centers such as Our World in Data (https://ourworldindata.org/coronavirus-source-data) and the Humanitarian Data Exchange (https://data.humdata.org/dataset/novel-coronavirus-2019-ncov-cases). We used data from the former site in this analysis.

**Variation by day of the week.** In the number of reported infections, a large variation has been observed by day of the week; reported infections tend to be fewer on the weekend than on the weekdays. There might have been variations in the original infectious activity due to human behavior, but it is more likely that this variation was caused by the delay in confirming infections and compiling the results at the weekend. The variation by day of the week is commonly observed, but there are large differences between countries, presumably due to the cultural difference in weekly activities (Fig 4).

Before analyzing a sequence of daily cases of a given country, we process the data as follows; we first obtained the gross daily variation $\beta_i$ in a week by averaging over the entire infection record (from March 1 2020 to the present), so that the average over a week is normalized as

$$\frac{1}{7} \sum_{\text{Sunday}}^{\text{Saturday}} \beta_i = 1. \tag{18}$$

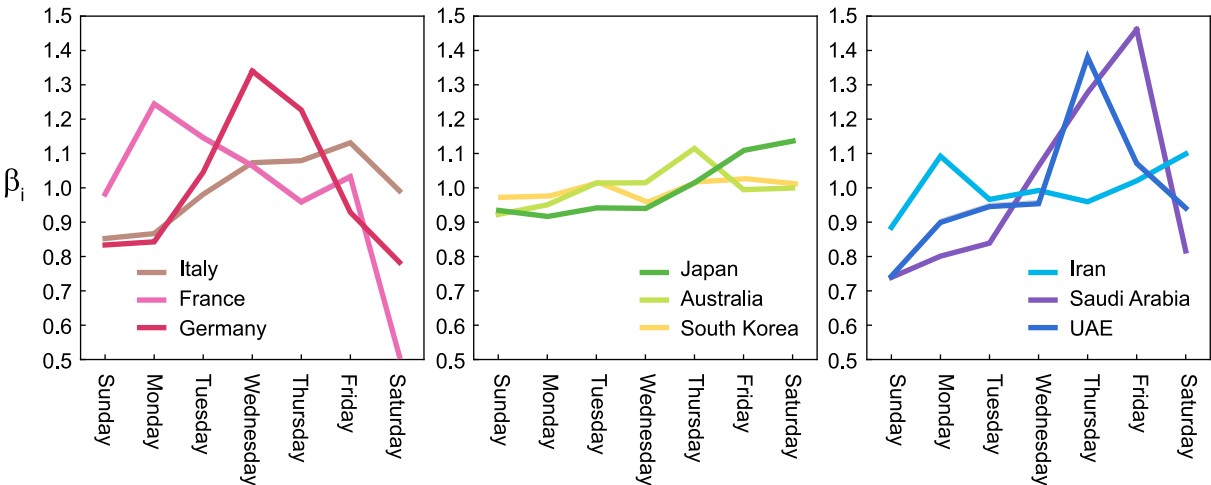

**Fig 4. Variations by day of the week in the number of reported infections $\{\beta_i\}$ computed for several countries.**

Then, we convert the original data of daily infections $\{n_1, \ldots, n_T\}$ to an adjusted dataset $\{v_1, \ldots, v_T\}$ by

$$v_i = n_i / \beta_i, \tag{19}$$

to which we apply the proposed state-space method.

**Diversity in the spread of the contagion.** In Fig 5, we show original daily new infection cases $\{n_1, \ldots, n_T\}$ and the adjusted dataset $\{v_1, \ldots, v_T\}$ of several countries. Below each panel of the daily cases, we demonstrate the reproduction number $\hat{R}_i$ estimated with the proposed state-space method:

Italy. A rapid increase in new cases was followed by a slow decrease. The estimated reproduction number was $\hat{R} > 1$ at the outset and dropped to $\hat{R} < 1$. It is interesting to note that the drop in the reproduction number occurred after political measures, such as lockdown and border closure, were enforced.

Japan. The number of cases is found to be relatively small compared to those in Europe. An increase in the number of cases was followed by a rapid decrease, and then by a second increase. Accordingly, the reproduction number exhibited a drop from $\hat{R} > 1$ to $\hat{R} < 1$, and then it increased to $\hat{R} > 1$.

Saudi Arabia. The number of new cases repeatedly moved up and down, and the estimated reproduction number $\hat{R}$ changed accordingly. It is observed that Ramadan has promoted increased reproduction number, as it may have facilitated human contact.

The United States. A rapid increase in new cases is followed by a very slow decrease, and then another growth. The estimated reproduction number $\hat{R}$ was higher than unity at the beginning, dropped off to near unity due to the confinement measures taken, but then it exceeded unity again. The political measures taken were found to vary by state, making it difficult to interpret the data from this country as a whole.

We also compared the proposed method with a conventional WT method [26] and EpiEstim [28] by applying them to these real data (Fig 5). It is also observed that the WT method and EpiEstim are easily influenced by the fluctuation of data. Results of other four countries are also shown in Fig 6.

**The reproduction number at the initial phase.** There have been debates about why infection rate and mortality rate change by orders of magnitude across different countries. Though these numbers likely reflect the confinement measures taken in individual countries, there might also have been differences across nations in susceptibility to COVID-19, reflecting not only genetic resistance but also lifestyle and cultural differences, such as shaking hands or hugging.

Because most governments did not implement serious confinement measures at the initial phase, the initial exponential increase of infections might reflect the natural susceptibility of citizens of each country. We realized that the estimated reproduction number was stable in a certain period before each country took confinement measures such as a lockdown or social distancing. Fig 7A depicts the reproduction numbers estimated with the proposed state-space

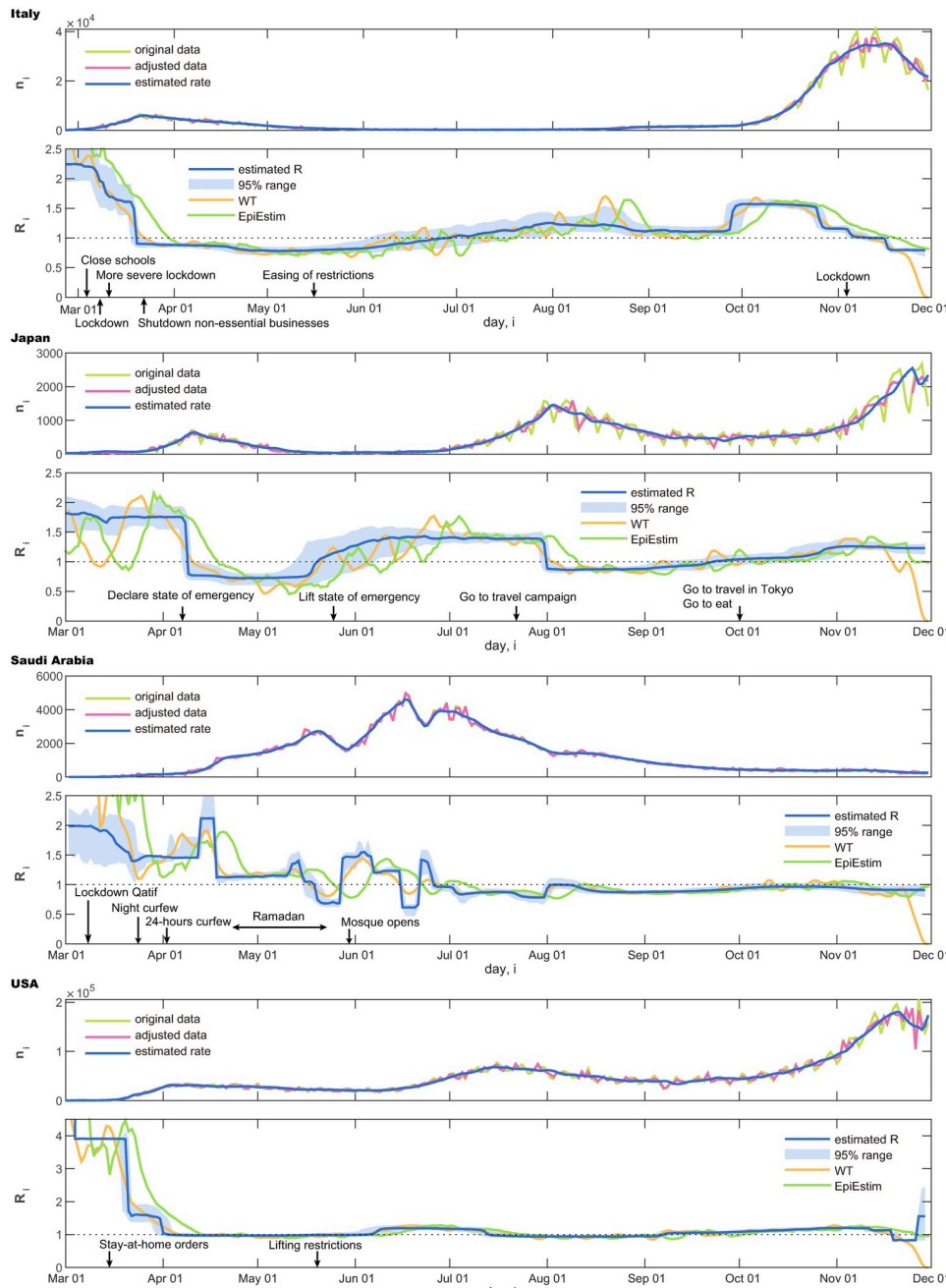

**Fig 5. Number of daily new cases and the reproduction number $\hat{R}_i$ estimated using the state-space method.** Italy, Japan, Saudi Arabia, and the USA.

method for 10 days until the day before the confinement measures of each country. The initial variation in the numbers of daily new cases is depicted in Fig 7B, indicating that the estimated reproduction number is correlated to the slope in the log plot. Here we have selected the period shifted by 5 days, by taking account of the typical transmission delays. We can observe that countries in different regions tend to cluster, indicating that the susceptibility tended to be similar between nations in the same region.

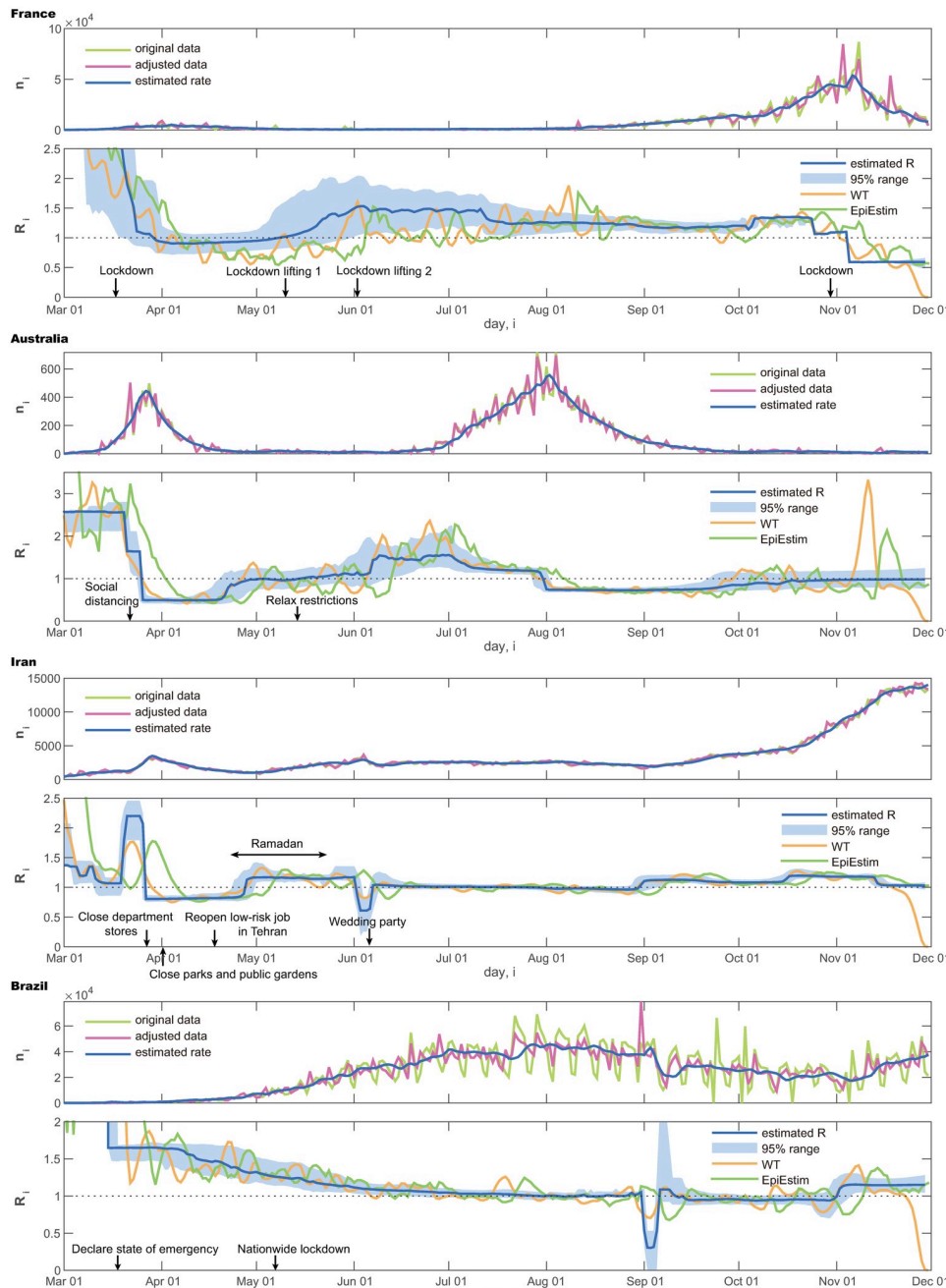

**Fig 6. Number of daily new cases and the reproduction number $\hat{R}_i$ estimated using thestate-space method.** France, Australia, Iran, and Brazil.

**The minimum reproduction number achieved in each country.** The degree of a drop in the estimated reproduction number could reflect the impact of non-pharmaceutical interventions such as a lockdown. It might be possible to quantify the effectiveness of political interventions in each country in terms of the relative percentage reduction in the reproduction number [13]. Here we searched for the minimum reproduction number averaged over 10 days that was achieved in each country. Fig 8A depicts the reproduction number for 10 days whose

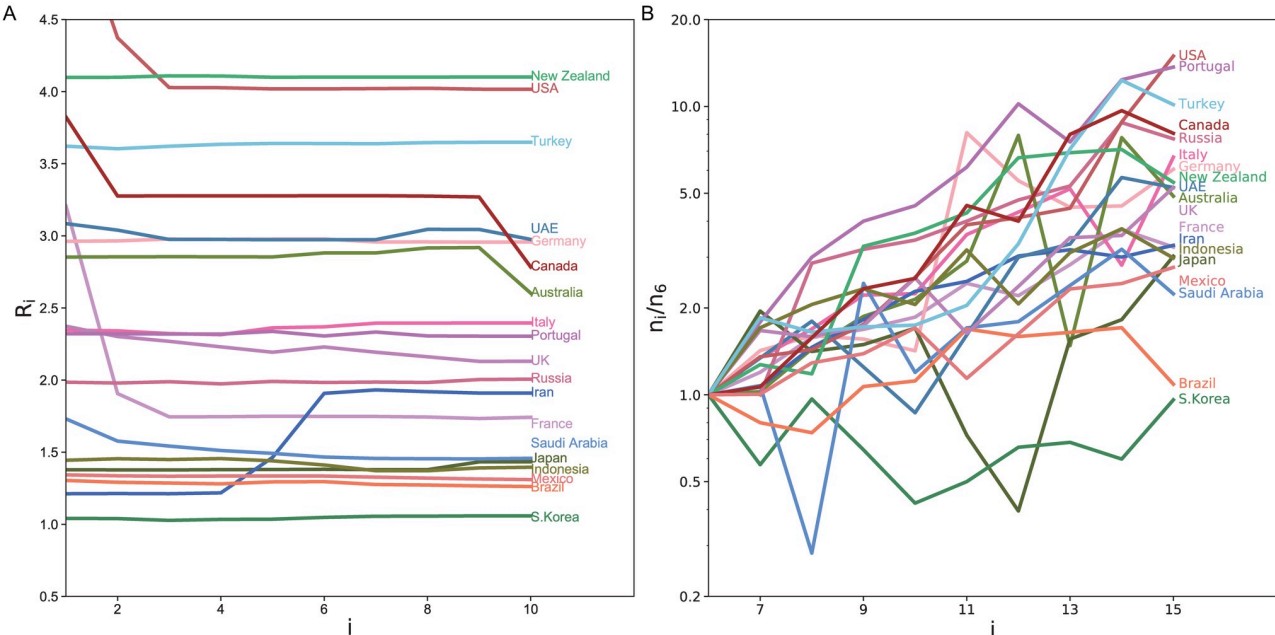

**Fig 7. The difference in the reproduction number at the initial phase.** (A) The reproduction numbers estimated with the proposed state-space method for 10 days until the day before the lockdown measures of each country. Days are counted from 12 days before the confinement measures. (B) Initial variation in the numbers of daily new cases; $n_i$ divided by $n_6$. The period is shifted by 5 days, by taking account of the typical transmission delay.

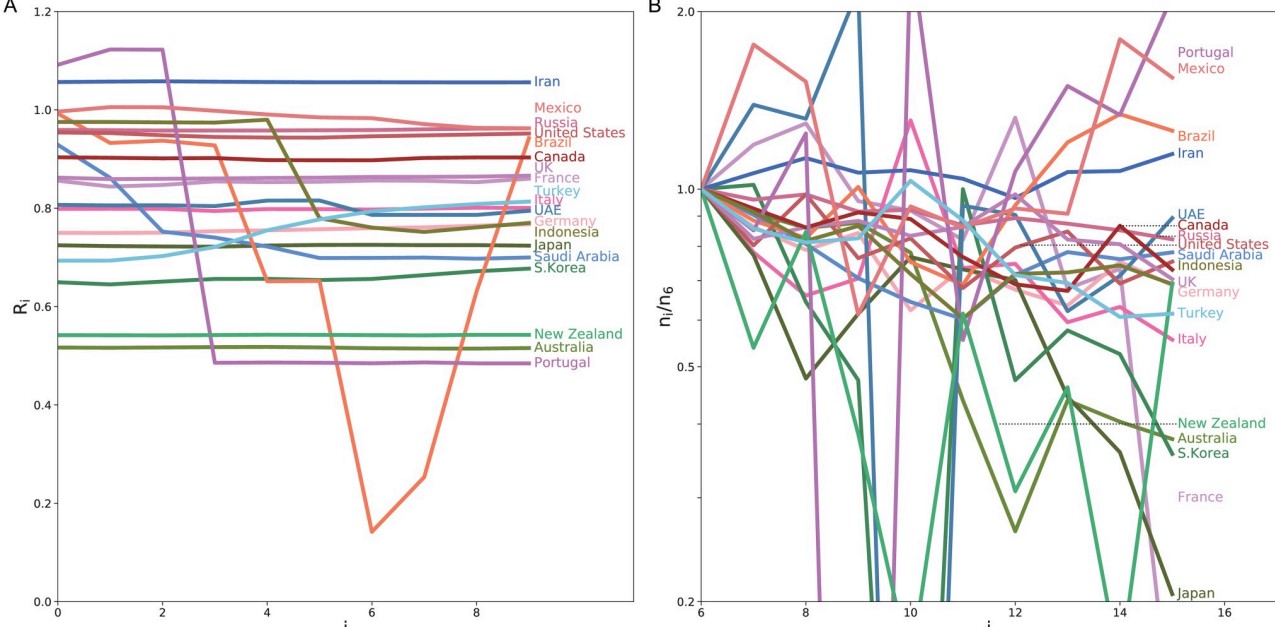

**Fig 8. The minimum reproduction number achieved in each country.** (A) The reproduction number for 10 days whose average takes minimum in each country. (B) Variation in the numbers of daily new cases; $n_i$ divided by $n_6$. The period is shifted by 5 days, by taking account of the typical transmission delay.

average takes minimum in each country. The variation in the numbers of daily new cases is depicted in Fig 8B, indicating that the estimated reproduction number is correlated to the slope in the log plot.

## Future prediction

Using the proposed method, it is also possible to predict the number of new cases in the future. This can be done by simulating the converted Hawkes process Eq (2) with the parameters estimated from the given data. One may adopt the reproduction number $R_i$ in near future as constant at the value of an endpoint of estimation if the current conditions are assumed to be maintained. Alternatively, one may also examine various time schedules of $R_i$, by assuming possible choices of relaxation or confinement measures.

In Fig 9 we applied the forecasting method to the data of Japanese daily cases. Assuming that we are on June 30, 2020, we have estimated the reproduction number $\hat{R}_i$ using the daily cases until that day. To predict the number of daily cases from July 1 to August 1, we ran the Hawkes process 100 times to obtain the expected daily cases. Firstly, we have assumed that the reproduction number remains the value obtained for the last day $R = 1.4$. Occasionally the reproduction number has not changed drastically in July, and accordingly, the predicted number of new cases is similar to the real data obtained in July.

We have also tested the cases in which the reproduction number is decreased to $R = 0.7$ due to confinement measures, or increased to $R = 1.8$ by liberalization. In this way, we may examine what might occur if political interventions are taken.

## Discussion

Society and the media currently alternate between hope and despair in response to the temporary decrease or increase of daily new COVID-19 infections, which came out after a long

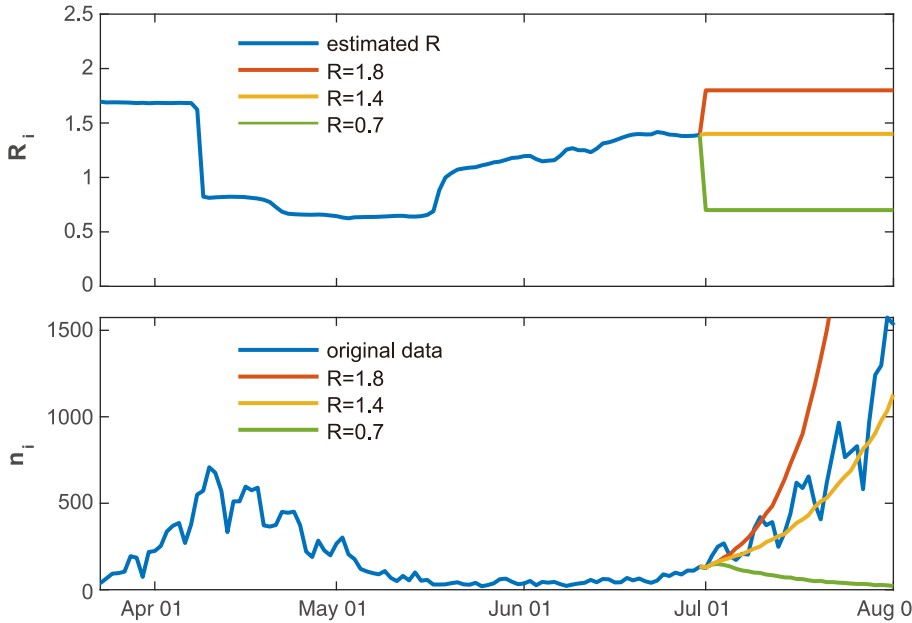

**Fig 9. Predicting the number of new cases in the future.** The forecasting method was applied to the data of Japanese daily cases, assuming that we were on June 30, 2020. We ran the Hawkes process 100 times to obtain the expected daily cases, by assuming that the reproduction number remains constant $R = 1.4$, which was obtained using the previous data (orange line). We also examined the cases in which the reproduction number is decreased to $R = 0.7$ due to confinement measures (green line) or increased to $R = 1.8$ by liberalization (red line).

latency period. To make an objective assessment of the current status, we have developed a state-space method for estimating the control status, in particular quantifying the time-dependent reproduction number $R$.

## Pros and cons

We have adopted the Hawkes process, or its discrete-time variant, in describing the variables underlying the transmission of disease. In contrast to ordinary differential equation models such as SIR or SEIR models, the Hawkes process is advantageous in that it explicitly specifies the distribution of transmission delays. However, the Hawkes process does not account for the finite size effect, in which infected and recovered people represent a finite fraction of the entire population. There have been some models that incorporate the finite population size effect into the Hawkes process, as has been done with the SIR or SEIR models [8, 29]. To analyze the current status of COVID-19, however, we do not take the finite size effect into account, as the fraction of the recovered or removed people is still less than a few % of the entire population.

We have converted the original Hawkes process Eq (1) into a discrete-time variant representing the expected number of events on a daily basis Eq (2) because the exact timing of infection event is not available for the case of COVID-19. It is noteworthy that Cheysson and Lang also developed a method for estimating parameters of the Hawkes process from counts data [30]. However, their method is based on a spectral likelihood, assuming stationarity in the underlying process. In contrast to this, we directly modeled the count time series and combined it with the state-space model to accommodate nonstationary data.

We introduced the Cauchy distribution Eq (6) into our analysis, assuming the stepwise changes in the reproduction number $R_i$. Accordingly, we were able to detect change-points from the posterior distribution taking on stepwise characteristics. As discussed by Kitagawa [21], the use of the heavy-tailed distribution enables us to express change-points, in contrast to using a Gaussian noise, which results in gentle changes. However, a drawback of the Cauchy distribution is that it causes slow convergence in the Monte Carlo simulation [31].

Interestingly, the drop in the reproduction number occurred after political measures, such as lockdown and border closure. It should be also noted that there may be an additional latency between the times at which political measures were conducted and the changes in the reproduction number, which may reflect the behavior change. This delay may also be country-specific. Therefore, it could be interesting to investigate the delay in the change-points in the reproduction number following social events.

When inferring the transmission of disease from daily confirmed cases, we have considered potentially erroneous observations made in the real data. We took into account counting errors by assuming a negative binomial distribution that represents the over-dispersion. We also took into account the variation by day of the week and adjusted the data by compensating for the periodic dependency. Note that there may still be an underestimation of infection numbers, as asymptomatic cases may have been overlooked. Though this is unavoidable unless the inspection is enforced, it is reported that the infections caused by asymptomatic people are relatively small (about 6%) for COVID-19 [19].

We have assumed that the transmission delay is a serial interval defined as the duration between symptom onsets of successive cases and adopted the log-normal distribution with the mean 4.7 days and SD 2.9 days, as suggested in reference [12]. As our mathematical formulation is general, it is possible to search for a more suitable transmission kernel $\phi_d$ without relying on such external knowledge, if the numbers of daily cases are accurately provided.

Here, we set the spontaneous occurrence rate to zero ($\mu' = 0$) in the analysis of real data. However, imported cases might be involved in the data. Also, we did not address censoring for

incomplete observation of the epidemic process in particular at the initial stage. These may cause bias in the estimations of the reproduction number at the early stage of the epidemic.

The most crucial assumption in the majority of mathematical model studies, including this study, is the mean-field assumption, in which all individuals are assumed to interact uniformly. Though difficult to incorporate, it is desirable to consider the heterogeneity of the real-world community in analyzing the communicability of disease.

Despite these assumptions, the proposed state-space method may be of worth in assessing the status of the disease systematically, based on reported daily confirmed cases. This method might serve as a reference for governments adopting variable regulations that should be changed according to current infection circumstances.

## Acknowledgments

We thank Shin Takagi, Masaki Ogura, Ryota Kobayashi, Takaaki Aoki, and Hideaki Shimazaki for their constructive comments on this manuscript, and Hidetaka Manabe for his technical assistance in developing a web-application program.

## Author Contributions

**Conceptualization:** Shinsuke Koyama, Shigeru Shinomoto.

**Data curation:** Shinsuke Koyama, Taiki Horie.

**Formal analysis:** Shinsuke Koyama.

**Funding acquisition:** Shigeru Shinomoto.

**Investigation:** Shinsuke Koyama, Shigeru Shinomoto.

**Methodology:** Shinsuke Koyama, Shigeru Shinomoto.

**Project administration:** Shigeru Shinomoto.

**Software:** Shinsuke Koyama, Taiki Horie.

**Visualization:** Shinsuke Koyama, Shigeru Shinomoto.

**Writing – original draft:** Shigeru Shinomoto.

**Writing – review & editing:** Shigeru Shinomoto.

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
