## [Decision Letter · Decision Letter 0]

5 Nov 2020

Dear Prof. Shinomoto,

Thank you very much for submitting your manuscript "Estimating the time-varying reproduction number of COVID-19 with a state-space method" for consideration at PLOS Computational Biology.

As with all papers reviewed by the journal, your manuscript was reviewed by members of the editorial board and by several independent reviewers. In light of the reviews (below this email), we would like to invite the resubmission of a significantly-revised version that takes into account the reviewers' comments.

We cannot make any decision about publication until we have seen the revised manuscript and your response to the reviewers' comments. Your revised manuscript is also likely to be sent to reviewers for further evaluation.

Sincerely,

Roger Dimitri Kouyos

Associate Editor

PLOS Computational Biology

Thomas Leitner

Deputy Editor

PLOS Computational Biology

Reviewer's Responses to Questions

**Comments to the Authors:**

Reviewer #1: This is a really interesting paper, and I am glad more work is being added applying Hawkes processes to Infectious diseases. The method is very nice, and i think novel to my knowledge. I do like the embedding of epidemiological approaches in state space. There are however some issues which means their results can be hard to justify. The use of case data when not linking cases to infections means the authors do not consider any incubation period or the possibility of asymptomatic transmission. This dilutes the validity of their reproduction number estimates. They really need to either connect R->infection->cases. I could not see the choice of phi being discussed. I assume its exponential, but if so this is wrong for infectious diseases. Hawkes processes are fundamentally linked to this phi, without it the model is incomplete.

I do comment the authors on their method, aside from a few missed details it is interesting and novel, but the could do much more to make their model more useful and accurate by considering the epidemiology. So the authors are best either focusing on a methods paper, and in which case a comparison to existing approachs (in addition to WT) would be well received, and of-course heavy caveating of the results. As an application paper, i would like a more careful consideration of the data.

I hope the authors will not begruge me, I realise these revisions would require quite a bit more work, but I believe they will strengthen the paper and ultimately help people using this in the future. However in general, I am very supportive of this work and belive it to be a good addition to the literature.

other comments

1) We have applied the state-space method <- this is introduced without me as the reader knowing what the authors are referring to

2) Limitations of using World in data should be added, factors like reporting accuracy and delay etc

3) It is interesting to note that the drop in the reproduction number occurred after political measures, such as lockdown and border closure, were enforced. <- A few points about why this might be the case would be good. Factors like slow adherence, but more plausibly that cases are lagged ahead of infections by an incubation period of 5 days or so. Obviously reporting is the biggest issue, testing was poor in most of europe in April.

4) A discussion of why the reproduction numbers differ at the start is good but i would like the authors to consider if its their modelling assumptions or data censoring rather than real effects.

5) The choice of kernel needs to be fully described in the main text. it underpins most of the dynamics

6) The sensitivity analysis of lambda should be presented in this paper

7) Posterior convergence and stats should be provided in the paper

8) Why is L 30

9) The choice of spontaneous occurenceis wrong, importation happened atleast for part of the period this paper considers

10) What is phi

minor comments which i leave to the authors to consider

1) The Kernel definition in (2) is nonstandard for a Hawkes process and can lead to the impression that its a discrete model

2) I do like the decomposition of the kernel into R and events, its novel

3) Might be worth mentioning the US for type C

4) The authors might consider looking at the HawkesN variant - they will find its like to SIR models interesting

Samir Bhatt

Reviewer #2: uploaded

**Have all data underlying the figures and results presented in the manuscript been provided?**

Reviewer #1: Yes

Reviewer #2: Yes

PLOS authors have the option to publish the peer review history of their article (what does this mean?). If published, this will include your full peer review and any attached files.

Reviewer #1: **Yes: **Samir Bhatt

Reviewer #2: No
---

## [Decision Letter · Decision Letter 1]

6 Jan 2021

Dear Prof. Shinomoto,

We are pleased to inform you that your manuscript 'Estimating the time-varying reproduction number of COVID-19 with a state-space method' has been provisionally accepted for publication in PLOS Computational Biology.

Best regards,

Roger Dimitri Kouyos

Associate Editor

PLOS Computational Biology

Thomas Leitner

Deputy Editor

PLOS Computational Biology

Reviewer's Responses to Questions

**Comments to the Authors:**

Reviewer #1: I am happy with the authors changes and their explanations. Thanks for their thoroughness.

**Have all data underlying the figures and results presented in the manuscript been provided?**

Reviewer #1: Yes

PLOS authors have the option to publish the peer review history of their article (what does this mean?). If published, this will include your full peer review and any attached files.

Reviewer #1: **Yes: **Samir Bhatt

---

## [Editor Report · Acceptance letter]

23 Jan 2021

PCOMPBIOL-D-20-01505R1 

Estimating the time-varying reproduction number of COVID-19 with a state-space method

Dear Dr Shinomoto,

I am pleased to inform you that your manuscript has been formally accepted for publication in PLOS Computational Biology. Your manuscript is now with our production department and you will be notified of the publication date in due course.

With kind regards,

Alice Ellingham
